# Sociodemographic Factors Associated with COVID-19 Vaccination among People in Guatemalan Municipalities

**DOI:** 10.3390/vaccines11040745

**Published:** 2023-03-28

**Authors:** Rewa Choudhary, Emily Carter, Jose Monzon, Allison Stewart, Jennifer Slotnick, Leslie L. Samayoa Jerez, David S. Rodriguez Araujo, Emily Zielinski-Gutierrez, Parminder S. Suchdev

**Affiliations:** 1Epidemic Intelligence Service, Centers for Disease Control and Prevention, Atlanta, GA 30329, USA; 2Division of Global Health Protection, Centers for Disease Control and Prevention, Atlanta, GA 30329, USA; 3Centers for Disease Control and Prevention Central America Regional Office, Guatemala City 01015, Guatemala; 4U.S. Agency for International Development (USAID), Guatemala City 01016, Guatemala; 5Ministerio Salud Publica y Asistencia Social, Guatemala City 01011, Guatemala; 6Council of Ministers of Health of Central America, San Salvador, El Salvador

**Keywords:** COVID-19 vaccination, Guatemala, equity

## Abstract

The Republic of Guatemala’s reported COVID-19 vaccination coverage is among the lowest in the Americas and there are limited studies describing the disparities in vaccine uptake within the country. We performed a cross-sectional ecological analysis using multi-level modeling to identify sociodemographic characteristics that were associated with low COVID-19 vaccination coverage among Guatemalan municipalities as of 30 November 2022. Municipalities with a higher proportion of people experiencing poverty (β = −0.25, 95% CI: −0.43–−0.07) had lower vaccination coverage. Municipalities with a higher proportion of people who had received at least a primary education (β = 0.74, 95% CI: 0.38–1.08), children (β = 1.07, 95% CI: 0.36–1.77), people aged 60 years and older (β = 2.94, 95% CI: 1.70–4.12), and testing for SARS-CoV-2 infection (β = 0.25, 95% CI: 0.14–0.36) had higher vaccination coverage. In the simplified multivariable model, these factors explained 59.4% of the variation in COVID-19 vaccination coverage. Poverty remained significantly associated with low COVID-19 vaccination coverage in two subanalyses restricting the data to the time period of the highest national COVID-19-related death rate and to COVID-19 vaccination coverage only among those aged 60 years or older. Poverty is a key factor associated with low COVID-19 vaccination and focusing public health interventions in municipalities most affected by poverty may help address COVID-19 vaccination and health disparities in Guatemala.

## 1. Introduction

Guatemala has the largest population among Central American countries (over 17 million) and is bordered by Mexico, Belize, Honduras, and El Salvador [1]. While considered an upper-middle-income country due to a GDP of $4603 per capita in 2020 [1], Guatemala has marked inequalities, with wealth held in a small sector of the population, low access to basic services for much of the population, and an overall lack of investment in the public sector [1,2,3].

Guatemala has the lowest COVID-19 vaccination coverage in Central America and is among the lowest compared with its regional neighbors in South America [4,5]. As of November 2022, there have been over one million SARS-CoV-2 cases and nearly 20,000 COVID-19-related deaths reported in Guatemala [6,7]. Within Guatemala, it was estimated that COVID-19-related mortality has been higher in people aged 60–69 years and in frontline healthcare workers [8]. As part of the National Vaccination Plan Against COVID-19, frontline healthcare workers were prioritized for vaccines when they initially became available in March 2021 [8,9]. In later phases, people aged 50 years and older and those with underlying medical conditions were prioritized [8,9]. Vaccines became available free of charge to the general public in May 2021 [8,9]. Vaccines were obtained through donations, via the COVID-19 Vaccines Global Access (COVAX) mechanism for which Guatemala was a “self-pay” country, and one bilateral purchase of the Sputnik vaccine [9]. To date, there are four COVID-19 vaccines available to Guatemalans and all of them are two-dose primary series courses (Pfizer/BioNTech, Oxford/AstraZeneca, Moderna, and Sputnik V) [10]. As of 30 November 2022, 19,960,793 COVID-19 vaccines had been administered, with 49.4% of the total population having received at least one dose, and approximately seven million people, or 40.0% of the total population, having completed a primary COVID-19 vaccination series with two doses [4,6,7,10]. Of those who had completed a primary vaccination series, 42.8% had received Moderna, 23.0% received Oxford/AstraZeneca, 18.4% received Sputnik V, and 15.9% had received Pfizer/BioNTech vaccines [11]. In addition, about four million people, or 24.0% of the population, had received one or two booster doses by 30 November 2022 [12].

The inequity in the global distribution of COVID-19 vaccines with preferential access for countries with higher per capita incomes and gross domestic products has been well-documented [13,14,15]. However, there are limited studies describing COVID-19 vaccination disparities within low- and middle-income countries, and many focus on vaccination intent. Studies that have explored sociodemographic factors associated with COVID-19 vaccination coverage have primarily focused on the United States. One survey conducted among U.S. adults found that participants with lower incomes, lower educational attainments, those without health insurance, who were non-Hispanic Black, and who lived outside of metropolitan areas had the lowest reported COVID-19 vaccination coverage and intent to get vaccinated [16]. Two other analyses showed that vaccination coverage was lower in rural compared with urban U.S. counties [17], and lower in counties with a higher percentage of people with incomes below the poverty threshold, experiencing unemployment, and not graduating from high school [18].

Several reasons have been posited for the low COVID-19 vaccination coverage in Guatemala. The country faces multiple challenges in its healthcare and public health system such as inadequate financing of the health sector, disparities in access to public health services in rural areas, and a shortage of healthcare workers [19,20]. COVID-19 vaccination coverage has been highest in the capital, Guatemala City, while rural areas with higher concentrations of Indigenous people have had lower vaccination coverage [5,21]. In the 2018 census, the Maya comprised 41.7% of the total population, and the Xinca were 1.8% of the total population [22]. The disproportionate burden of COVID-19 on Indigenous people and those of low socioeconomic status has been studied regionally, for instance, in Colombia [23]. A 2021 UNESCO report on COVID-19 vaccination in Latin America and the Caribbean noted that in communities with higher “unemployment or informal employment, or where ethnic groups live, there is a higher prevalence of COVID-19 and a higher risk of mortality” [24]. Moreover, there is evidence of vaccine hesitancy among people with lower levels of institutional trust, those living in rural areas, and those experiencing economic insecurity [25,26,27]. Early vaccination outreach in Guatemala was often conducted in Spanish using mainstream media instead of through local organizations, and using local Indigenous languages, according to a Pan American Health Organization report [5]. As part of the Ministry of Public Health and Social Assistance (MSPAS) *Strategy to Strengthen the COVID-19 Vaccination Plan in Rural Areas*, vaccination activities have more recently incorporated community leader guidance and local media campaigns [28].

Given the limited studies describing COVID-19 vaccination coverage disparities within Guatemala, we performed an ecological analysis to understand the association between sociodemographic factors and primary vaccination coverage among the 340 Guatemalan municipalities. Identifying factors associated with low vaccination could inform strategies to improve COVID-19 vaccine coverage and other public health interventions in Guatemala.

## 2. Materials and Methods

We performed a cross-sectional analysis of aggregated COVID-19 vaccination coverage data in Guatemala from 13 February 2020 to 30 November 2022. As the lowest level of data availability was at the municipal level, we conducted an ecological analysis of factors associated with primary COVID-19 vaccine series’ coverage by municipality.

Sociodemographic data variables at the municipal level were obtained through the 2018 Guatemala Population and Housing Census (Table 1) [22]. Municipalities are organized into 22 departments, and some variables only available at the departmental level were obtained through the 2014–2015 Demographic and Health Survey [29]. We chose sociodemographic variables *a priori* that could be proxies for healthcare access, poverty, and related variables that we hypothesized to be related to vaccination uptake [30,31,32]. We elected to use the general poverty indicator (percentage of each municipality population experiencing poverty) developed by Figueroa Chávez and colleagues and shared with our team [33]. Figueroa Chávez and colleagues developed the general poverty indicator by using the associations between sociodemographic variables in the 2018 Census and the poverty measure in the 2014 National Survey of Living Conditions [34] to estimate poverty at the municipality level [35]. According to their findings, the general poverty indicator ranged from 10.56% in the Jocotenango municipality in Sacatepéquez, to 94.59% in the Senahú municipality in Alta Verapaz [35]. COVID-19 vaccination data differentiated by municipality, department, and age were available at the MSPAS of Guatemala surveillance websites from 25 February 2021 to 30 November 2022 [11,12]. SARS-CoV-2 testing data (either antigen or polymerase chain reaction tests) and death data were also available through MSPAS from 13 February 2020 to 30 November 2022 [11]. A completed primary COVID-19 vaccination course was considered to be two doses of any of the four nationally available vaccines among people aged six years and older, consistent with current national guidelines [11]. The data used in this study were all de-identified, aggregated, and, with the exception of the general poverty indicator, publicly available. 

Municipal-level independent variables included in the model were the percentage of each municipality population reported to be of Mayan ethnicity, living in a rural residence, of the female sex, having attained primary school or higher educational level, experiencing poverty, aged 0–17 years or ≥60 years, and having died due to COVID-19 (Table 1). The reported number of SARS-CoV-2 tests by municipality was an additional independent variable. Independent variables at the departmental level included the under-five childhood mortality rate, the percentage of women aged 15–49 years who reported problems accessing health services when ill due to distance to a health establishment, the percentage of children aged 12–23 months who had received a third dose of Pentavalent vaccine (a combination vaccine against diphtheria, tetanus, pertussis, hepatitis B, and *Haemophilus influenzae* type b), and the Gini coefficient indicating income inequality. The dependent variable was the percent coverage of each municipality population with a complete primary COVID-19 vaccination course. Proportions of municipalities with completed COVID-19 vaccination and SARS-CoV-2 tests exceeding 100.0% (as total population estimates were from 2018) were capped at 99.0%. Given that the proportion of the population that died from COVID-19 by municipality was relatively small, the variable was scaled by 100 in the model to achieve a similar order of magnitude to the other variables.

Two subanalyses were also performed to assess whether the demographic associations with vaccination in the overall model were consistent among the subgroups. In the first, the dependent variable was limited to the population aged 60 years or older who had completed COVID-19 vaccination. We chose this subgroup given the initial national focus on vaccinating older adults. In the second subanalysis, the SARS-CoV-2 cases and COVID-19 vaccination data were confined to the period of the highest national COVID-19-related death rate, from 13 February 2020 to 1 October 2021 [6]. All count variables (derived from the census and the MSPAS) were converted to percentages to account for differences in municipal total populations. Data on deaths due to COVID-19 were missing for four municipalities (San Juan Tecuaco, Santa Rosa; Concepción, Sololá; Santa Catarina Palopó, Sololá; Río Blanco, San Marcos) and were removed from the multivariable models. 

We calculated descriptive statistics for sociodemographic characteristics among municipalities and departments. We used Pearson correlation coefficients and variance inflation factors to assess potential collinearity within our model. The poverty indicator used in our analysis was developed using some of the variables included in our model, however, these common variables were not heavily weighted in the poverty index [33,35]. As this was the most robust measure of poverty by municipality despite potential collinearity, we performed a sensitivity analysis of the model without the poverty indicator and found similar results and chose to retain this variable (Appendix A). We identified municipalities with high Indigenous populations, rurality, and poverty who achieved a COVID-19 vaccination coverage of at least 70%, according to World Health Organization (WHO) guidelines [36]. We assessed relationships between municipal- and departmental-level factors and COVID-19 vaccination using multi-level modeling, allowing for random department-level intercepts to account for differences between departments. The model results were robust to different specifications of the underlying error distribution. A multi-level linear regression model was selected to maximize both model fit and interpretability. All variables were included in the full multivariable model, and those variables with associations significant at *p* < 0.05 were included in the simplified multivariable model. We used a normal approximation of a 1000 replicate parametric bootstrap to generate our 95% confidence intervals and present both the marginal R^2^ (representing the proportion of the variance explained by the model-fixed effects) and conditional R^2^ (representing the proportion of the variance explained by both the fixed and random effects) for each model. All analyses were performed using R Statistical Software (v.4.2.2; Vienna, Austria) [37]. We hypothesized, based on our literature review, that COVID-19 vaccination would be negatively associated with higher poverty, rurality, and Indigenous population. 

## 3. Results

### 3.1. Demographic Data

Among the 340 municipalities, as of 30 November 2022, 7.05 million persons had completed a primary COVID-19 vaccine course. The median proportion of vaccination coverage among people aged at least six years in the municipalities was 42.3% (interquartile range, IQR, 31.8–53.8%) (Table 2). The median proportion of the female sex among municipalities was 51.3% (IQR 50.6–52.2%). The median proportion of the population aged 60 years or older among all municipalities was 7.8% (IQR 6.9–9.3%), and the median proportion of those identifying as Maya was 30.0% (IQR 2.9–91.4%). Four out of 151 majority Maya municipalities, San José Chacayá, Santa Catarina Barahona, Chimaltenango, and San Lorenzo, had at least 70% of their populations who had completed COVID-19 vaccination, and this relationship was inversely correlated (ρ = −0.299, *p* < 0.001) (Figure 1A). Guatemala, Mixco, Jocotenango, and San Lucas Sacatepéquez municipalities had the highest proportion of their populations (≥40%) who had received at least a primary school education. The median proportion of rural residence among all municipalities was 64.8% (IQR 37.1–82.7%), with an inverse correlation (ρ = −0.417, *p* < 0.001) with COVID-19 vaccination coverage (Figure 1B). The median proportion of people experiencing poverty among the municipalities was 60.8% (IQR 43.8–75.8%), with an inverse correlation with COVID-19 vaccination (ρ = −0.634, *p* < 0.001) (Figure 1C). Overall, the municipalities with the highest COVID-19 vaccination coverage were Guatemala, San José, and San José del Golfo. By November 2022, 23 municipalities had completed primary series vaccination in at least 70% of their populations (Figure 2). Additional summarizing demographic data among the municipalities are shown in Table 2.

### 3.2. Factors Associated with COVID-19 Vaccination

In the bivariate analysis of all COVID-19 vaccination data as of 30 November 2022, significant factors (α = 0.05) negatively associated with primary COVID-19 vaccination course coverage by municipality, adjusting for departmental level differences, included the proportion of the municipality identifying as Mayan (β = −0.15, 95% CI: −0.21–−0.10), the proportion of the municipality living in a rural residence (β = −0.19, 95% CI: −0.25–−0.13), the proportion of the municipality experiencing poverty (β = −0.54, 95% CI: −0.63–−0.46), the proportion of the municipality in the 0–17 years age group (β = −1.81, 95% CI: −2.16–−1.47), departmental-level under-five childhood mortality rate (β = −0.09, 95% CI: −0.46–0.27), the proportion of those in the department reporting difficulty accessing healthcare due to distance from a health facility (β = −0.82, 95% CI: −1.12–−0.53), and the department’s Gini coefficient (β = −0.62, 95% CI: −1.20–−0.08) (Table 3). 

Factors positively associated with primary COVID-19 vaccination series coverage by municipality, adjusting for departmental level differences, included the proportion of the municipality with at least a primary school education (β = 1.40, 95% CI: 1.18–1.62), the proportion of female sex in the municipality (β = 2.53, 95% CI: 1.02–4.03), proportion of the municipality in the 60 years and older age group (β = 4.54, 95% CI: 3.57–5.49), proportion of the municipality tested for SARS-CoV-2 infection (β = 0.52, 95% CI: 0.43–0.61), reported deaths due to COVID-19 in the municipality (β = 1.17, 95% CI: 0.94–1.41), and proportion of 12–23 months old children in the department who had received the third Pentavalent vaccine (β = 0.65, 95% CI: 0.20–1.11). 

After adjusting for all covariates and departmental effects in the full model, the proportions of the municipal population who (1) had received at least a primary school education, (2) were experiencing poverty, (3) were below the age of 18 years, (4) were aged 60 years and above, and (5) tested for SARS-CoV-2 remained significantly associated with complete vaccination coverage (Table 3, Section “Full multivariable model”). In the simplified multivariable model (Table 3, Section “Simplified multivariable model”), when adjusting for covariates and departmental level differences, a 10% higher proportion of people experiencing poverty within a municipality was associated with 2.5% lower COVID-19 vaccination coverage (95% CI: −4.33–−0.70). Conversely, a 10% increase in the proportion of the municipality having received at least a primary school education was associated with 7.4% higher COVID-19 vaccination coverage (95% CI: 3.83–10.75); a 10% increase in the proportion of the municipality aged <18 years was associated with 10.7% higher COVID-19 vaccination coverage (95% CI: 3.55–17.67); a 10% increase in the proportion of the municipality in the 60 years and older age group was associated with 29.4% higher COVID-19 vaccination coverage (95% CI: 17.00–41.21); and a 10% higher proportion of the municipality tested for SARS-CoV-2 infection was associated with a 2.5% higher COVID-19 vaccination coverage (95% CI: 1.37–3.55). Overall, the marginal R^2^ value was 0.496, and the conditional R^2^ value accounting for the covariates and departmental-level differences was 0.594.

### 3.3. Factors Associated with COVID-19 Vaccination among People Aged 60 Years or Older

Nationally, 61.9% of those aged 60 years or older had completed COVID-19 vaccination by 30 November 2022 [12]. In the analysis of this subgroup, significant factors (α = 0.05) in the bivariate model negatively associated with COVID-19 vaccination, adjusting for departmental level differences, were similar to the overall analysis and included the proportion of the municipality identifying as Mayan, the proportion living in a rural residence, and the proportion experiencing poverty, but did not include the departmental-level under-five childhood mortality rate and the department’s Gini coefficient (Table 4). Factors positively associated in the bivariate model were the same as in the overall analysis.

After adjusting for all covariates and departmental effects in the full model, the proportions of the municipal population (1) living in a rural residence, (2) having received at least a primary school education, (3) experiencing poverty, (4) of female sex and (5) tested for SARS-CoV-2 remained significantly associated with complete vaccination coverage (Table 4, Section “Full multivariable model”). In the simplified multivariable model (Table 4, Section “Simplified multivariable model”), when adjusting for covariates and departmental level differences, a 10% higher proportion of people experiencing poverty within a municipality was associated with 2.0% lower complete COVID-19 vaccination coverage (95% CI: −3.79–−0.28). A 10% increase in the proportion of the municipality living in a rural residence was associated with a 0.9% higher COVID-19 vaccination coverage (95% CI: 0.27–1.59); having received at least a primary school education was associated with 9.8% higher COVID-19 vaccination coverage (95% CI: 5.66–13.72); a 10% increase in the proportion of the municipality of female sex was associated with 20.6% higher COVID-19 vaccination coverage (95% CI: 6.39–34.51); and a 10% higher proportion of the municipality tested for SARS-CoV-2 infection was associated with a 3.1% higher COVID-19 vaccination coverage (95% CI: 1.90–4.37). The marginal R^2^ value was 0.487, and the conditional R^2^ value accounting for the covariates and departmental-level differences was 0.600.

### 3.4. Factors Associated with COVID-19 Vaccination up to 1 October 2021

As of 1 October 2021, immediately after the peak of COVID-19-related deaths in Guatemala, 2.58 million persons (15.1% of the total population) had completed a primary COVID-19 vaccine course [11]. In the bivariate analysis of all COVID-19 vaccination data up to 1 October 2021, significant factors negatively associated with COVID-19 vaccination coverage by municipality (α = 0.05), adjusting for departmental level differences, were similar to the overall analysis and included the proportion of the municipality identifying as Mayan, the proportion living in a rural residence, and proportion experiencing poverty, but did not include the departmental-level under-five childhood mortality rate and the department’s Gini coefficient (Table 5). Factors positively associated in the bivariate model were the same as in the overall analysis. 

After adjusting for all covariates and departmental effects in the full model, the proportion of the municipal population (1) having received at least a primary school education, (2) experiencing poverty, (3) aged 60 and above, and (4) tested for SARS-CoV-2 remained significantly associated with complete vaccination coverage (Table 5, Section “Full multivariable model”). In the simplified multivariable model (Table 5, Section “Simplified multivariable model”), when adjusting for covariates and departmental level differences, a 10% higher proportion of people experiencing poverty within a municipality was associated with 1.1% lower COVID-19 vaccination coverage (95% CI: −1.90–−0.39); a 10% increase in the proportion of the municipality having received at least a primary school education was associated with 1.6% higher COVID-19 vaccination coverage (95% CI: −0.12–3.28); a 10% increase in the proportion of the municipality in the 60 years and older age group was associated with 13.9% higher COVID-19 vaccination coverage (95% CI: 8.93–18.73); and a 10% higher proportion of the municipality tested for SARS-CoV-2 infection was associated with a 2.6% higher COVID-19 vaccination coverage (95% CI: 1.46–3.80) (conditional R^2^ = 0.615).

## 4. Discussion

In this cross-sectional analysis of COVID-19 vaccination coverage among Guatemalan municipalities, we provide population-level information on sociodemographic and health system variables associated with vaccination. In the adjusted multi-level model evaluating vaccination data as of 30 November 2022, municipalities with higher proportions of people experiencing poverty had lower COVID-19 vaccination coverage. Municipalities with higher proportions of people who had received at least a primary school education, children, people aged 60 years or older, and testing for SARS-CoV-2 infection had higher COVID-19 vaccination coverage. In our subanalyses, the timing of the country’s response to the pandemic did not appear to notably affect the results, as factors associated with vaccination coverage in the overall model were similar to the point at which Guatemala had just passed its highest daily death rate. Additionally, poverty, educational level, and prevalence of testing for SARS-CoV-2 infection remained significant factors associated with COVID-19 vaccination coverage among Guatemalans aged ≥60 years.

Our findings are generally consistent with those from previous studies. Prior studies have largely focused on factors related to the intent to vaccinate rather than the completion of COVID-19 vaccination. A 2022 global, population-based analysis noted that participants identifying as female, in older age groups, with a higher level of education, and with health insurance reported being more willing to get vaccinated [32]. In a study of Latin American and Caribbean countries, those with a university education, residence in an urban area, and a higher perceived likelihood of contracting COVID-19 had higher intentions to be vaccinated [31]. With regards to age, our model showed that municipalities with more children and people aged 60 years or older had higher vaccine coverage when adjusting for other covariates and variations at the departmental level. This is expected given that older populations were prioritized under national vaccination planning given their elevated risk of severe COVID-19 [8]. The role of children is less clear, especially as vaccines for children under 12 years of age only became available in 2022, and there are no vaccines for children under six years of age at the time of analysis [38]. Possibly, concern over the well-being of children motivated parents’ vaccination, or that when vaccines were available for children, other family members were also vaccinated. The role of children in the community could be an area of further investigation. 

It is also expected that municipalities with more SARS-CoV-2 testing had a higher proportion of the population that was vaccinated for COVID-19 as these municipalities may have more access to health facilities or services. The MSPAS provided free testing to people with symptoms or to those who were COVID-19 contacts, however, these services were generally offered at health facilities that were not always accessible to rural populations [9,21,39]. It is possible that the presence of more testing resources could positively influence individuals to receive COVID-19 vaccinations. This would support public health interventions, such as making SARS-CoV-2 testing more accessible in rural areas through mobile health units. In our model, higher proportions of deaths due to COVID-19 in the municipalities were not significantly associated with increased vaccination. While Guatemala has reported less excess mortality compared with other countries within the region [40], the excess mortality was found to be 46% higher compared with confirmed COVID-19 death counts, according to a study by Martinez-Folgar and colleagues [41], indicating that mortality and case estimates are likely underestimated, which may have affected our analysis. Moreover, their study showed that most deaths appeared to occur at home, further highlighting barriers to healthcare access that are likely reflected in low COVID-19 vaccination coverage and possibly higher mortality. 

In unadjusted models, we observed significant associations between lower COVID-19 vaccination coverage and Indigenous identity, rural residence, poverty, and self-reported difficulty accessing healthcare. In the adjusted model, of these sociodemographic variables, only the proportion of the municipality experiencing poverty remained negatively associated with vaccination coverage. It is possible that poverty partially explains the observed associations between COVID-19 vaccination and other sociodemographic variables, such as Indigenous identity, rurality, and healthcare access. We found a few municipalities with high rurality and Indigenous populations that reached at least 70% vaccination coverage. There were only three municipalities with 50% or more of their population experiencing poverty that reached 70% vaccination coverage. Poverty remained significantly associated with low vaccination coverage when the analysis was restricted to the time that COVID-19 deaths peaked, or to coverage among those aged 60 years or older. Even as the risk of mortality due to COVID-19 has been shown to be higher among those in the lowest socioeconomic strata [23], there is evidence that economic insecurity was associated with fear of adverse effects from the vaccine in Latin America [27]. Additionally, the monetary and opportunity costs of accessing vaccination sites, missing work, arranging childcare, etc., have been described as potential barriers to vaccine access [5,21]. Therefore, while Indigenous and rural communities are at a higher risk for low vaccine access, it may be particularly effective to use poverty indices when designing community-wide vaccination interventions in Guatemalan municipalities, and to focus on interventions such as transportation, childcare, and alternative hours of service that can overcome cost-related barriers. Further, research to better understand the structural determinants of poverty, including class, gender, and race, can help guide future interventions [42]. Additional research on the monetary and opportunity costs of accessing vaccination within Guatemala may be needed. Lastly, outreach specifically to areas with lower access to primary school education and vaccination programming that accommodates potential literacy issues may be considered.

There are limitations to this study which should be considered. Our analysis was conducted at the municipal level as we did not have access to community estimates or individual-level data that could possibly provide more complex explanations of low vaccination coverage among certain sociodemographic groups. Our conclusions at the population level may not be applicable to specific sociodemographic groups within municipalities. Secondly, factors such as poverty, Indigenous identity, and rurality are complex and interrelated, and it is difficult to assess their relationship to vaccination and healthcare access in isolation. The proxies we used for healthcare access, such as testing for SARS-CoV-2 infections and vaccination program reach, such as childhood Pentavalent vaccination coverage, may not capture the intricacies of the political, economic, and historical reasons for low COVID-19 vaccination coverage. Additionally, our analysis may differ depending on alternative definitions of vaccination coverage, such as partial vaccination with one dose or coverage with booster doses. Lastly, as we relied on data from the most recent national census, some of our findings may not reflect the population during the COVID-19 pandemic.

## 5. Conclusions

Through our multi-level modeling approach, we were able to identify sociodemographic factors associated with COVID-19 vaccination at the municipality level. Our findings show more granularly where COVID-19 vaccination is lagging in Guatemala and which municipalities could benefit from more focused vaccination activities. Municipalities with populations experiencing higher poverty had lower vaccination coverage, and municipalities with higher proportions of primary education completion, children, people aged 60 years and older, and more testing for SARS-CoV-2 infection had higher vaccination coverage. COVID-19 vaccine delivery and public health outreach may be focused on communities experiencing more poverty. While there has historically been a difficulty with healthcare delivery to communities experiencing poverty, interventions based on poverty indices may help mitigate the effects of the COVID-19 pandemic on such communities and ultimately improve health equity. 

## Figures and Tables

**Figure 1 vaccines-11-00745-f001:**
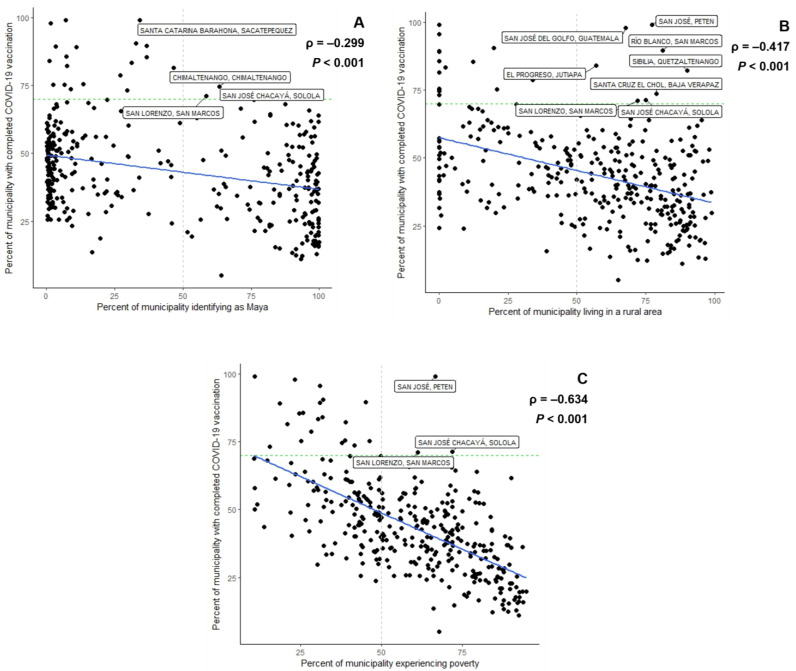
Guatemalan municipalities by percent of primary COVID-19 vaccination series coverage versus (**A**) percent of municipality population of Mayan identity, (**B**) percent of municipality population living in a rural area, and (**C**) percent of municipality population experiencing poverty. Labeled municipalities are those with >70% completed COVID-19 vaccination (green dashed line) and >50% of the Mayan population (**A**), rural residence (**B**), or people experiencing poverty (**C**) (gray dashed line).

**Figure 2 vaccines-11-00745-f002:**
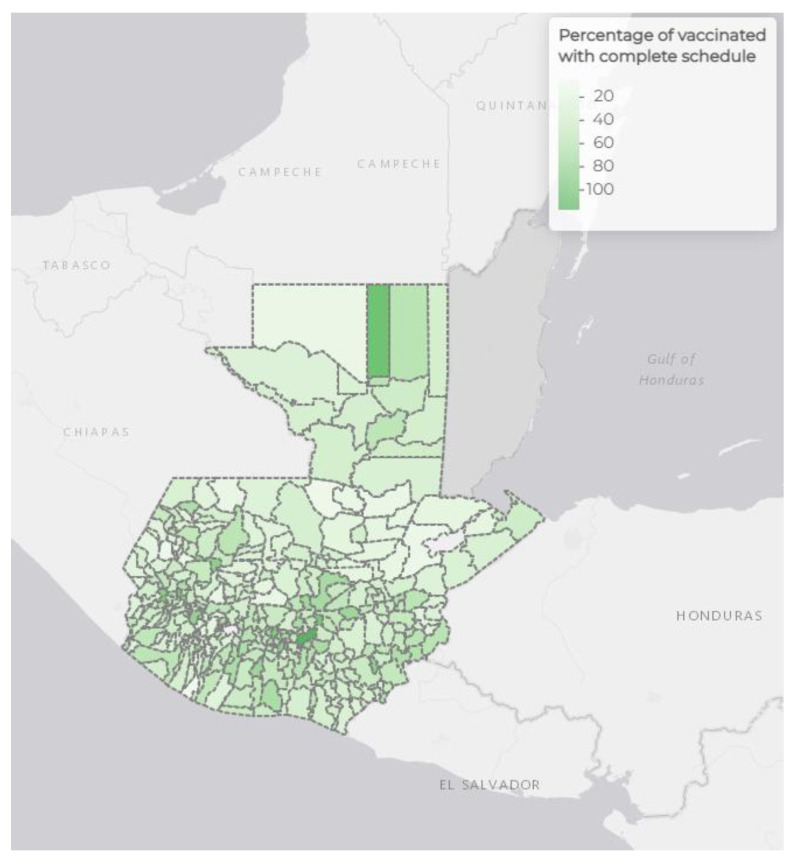
Primary COVID-19 vaccination series coverage by municipality in Guatemala. Image adapted from Guatemala Ministry of Health and Social Assistance, 25 February 2021 to 30 November 2022 [11].

**Table 1 vaccines-11-00745-t001:** Data elements used in the analysis.

Variable	Source
Total municipality population	2018 Guatemala Population and Housing Census
Female sex population in the municipality	2018 Guatemala Population and Housing Census
Population aged 0–17 years in municipality	2018 Guatemala Population and Housing Census
Population aged 18–59 years in municipality	2018 Guatemala Population and Housing Census
Population aged 60 years or older in municipality	2018 Guatemala Population and Housing Census
Ethnicity identification in the municipality	2018 Guatemala Population and Housing Census
Population in a municipality having received at least primary school education	2018 Guatemala Population and Housing Census
Population in municipality with household in a rural location	2018 Guatemala Population and Housing Census
Population in municipality experiencing poverty	Figueroa Chavez et al., 2020 [30]
Department-level childhood mortality rate (deaths per 1000 livebirths) among children aged under five years	2014–2015 Demographic and Health Survey
Percentage of women aged 15–49 years in department who reported having problems accessing health services when ill due to distance to a health establishment	2014–2015 Demographic and Health Survey
Percent of children aged 12–23 months in the department who have received the third Pentavalent vaccine dose	2014–2015 Demographic and Health Survey
Department Gini coefficient (%)	2014–2015 Demographic and Health Survey
SARS-CoV-2 vaccination status among the municipality population aged six years or older (incomplete, complete, one booster dose, two booster doses)	Ministry of Public Health and Social Assistance (MSPAS) of Guatemala, 2021–2022
Municipality population aged 60 years or older with completed SARS-CoV-2 primary vaccination course	Ministry of Public Health and Social Assistance (MSPAS) of Guatemala, 2021–2022
SARS-CoV-2 tests reported per municipal population.	Ministry of Public Health and Social Assistance (MSPAS) of Guatemala, 2020–2022
Deaths due to COVID-19 among the municipality population	Ministry of Public Health and Social Assistance (MSPAS) of Guatemala, 2020–2022

**Table 2 vaccines-11-00745-t002:** Demographic characteristics among the 340 municipalities and 22 departments in Guatemala. Data source: Guatemala Population and Housing Census 2018, 2014–2015 Demographic and Health Survey, and Guatemala Ministry of Health and Social Assistance. The poverty variable is a model estimate from Figueroa Chávez et al. (2020) [33,35]. SARS-CoV-2 cases and vaccination data are from 13 February 2020 to 30 November 2022, except where indicated.

Municipality Characteristic	N Median (IQR)	% ^a^ Median (IQR)
Population	28,156.5 (15,730.8–51,426.0)	-
Female sex	14,580.0 (8067.3–27,052.3)	51.3 (50.6–52.2)
Age group (years)		
0–17	11,599.5 (6227.3–21,980.8)	40.8 (37.4–45.3)
18–59	14,131.0 (7891.3–25,263.5)	51.1 (48.0–53.4)
≥60	2226.0 (1321.0–3797.8)	7.8 (6.9–9.3)
Ethnicity		
Maya	7129.0 (1008.8–25,847.5)	30.0 (2.9–91.4)
Garifuna	25.0 (13.0–50.8)	0.1 (0.1–0.1)
Xinka	4.0 (1.0–16.0)	0.0 (0.0–0.1)
Latino(a)	11,362.5 (2463.0–26,008.5)	63.7 (8.2–93.5)
Educational level primary school and above	18,065.0 (10,058.5–32,712.8)	74.4 (68.5–78.9)
Household rural location	13,458.5 (6260.5–28,052.5)	64.8 (37.1–82.7)
People experiencing poverty	16,086.0 (7909.0–30,343.5)	60.8 (43.8–75.8)
SARS-CoV-2 vaccination status		
Vaccine eligible population (≥6 years)	27,156.0 (15,180.3–50,978.5)	-
Incomplete	13,853.5 (7890.8–23,373.3)	53.0 (43.0–66.6)
Complete	10,633.5 (6286.3–18,466.8)	42.3 (31.8–53.8)
One booster dose	4813.0 (2887.5–8097.5)	18.8 (13.1–27.0)
Two booster doses	385.5 (183.8–952.0)	1.7 (0.9–3.0)
MSPAS ^b^ SARS-CoV-2 indicators		
Confirmed cases	917.5 (442.0–1899.8)	3.0 (1.7–5.4)
Tests reported	5248.5 (2798.0–10,824.5)	19.4 (10.7–28.6)
Deaths due to COVID-19	20.0 (10.0–38.0)	0.07 (0.0–0.1)
People aged 60 or more years with complete vaccination	1173.5 (733.8–2071.0)	53.0 (41.5–66.9)
Measures as of 1 October 2021		
Complete vaccination	3160.5 (1918.0–5897.8)	13.1 (8.5–19.6)
Tests reported	2129.0 (966.5–4492.8)	7.9 (4.5–12.4)
Deaths due to COVID-19	16.5 (8.0–30.8)	5.6 (3.2–9.5)
**Departmental Characteristic**		**%** **Median (IQR)**
Under-5 childhood mortality rate (deaths per 1000 live births)		37.0 (31.0–42.5)
Difficult access to healthcare facilities due to distance		38.6 (33.8–46.9)
12–23 months old children receiving third Pentavalent vaccine		86.9 (82.0–90.4)
Gini coefficient		30.0 (30.0–40.0)

^a^ Proportion derived by dividing by total population as provided by respective data source. ^b^ Ministry of Public Health and Social Assistance.

**Table 3 vaccines-11-00745-t003:** Association between sociodemographic factors (by municipalities and departments) and two-dose vaccination coverage (%) by municipalities in Guatemala (N = 336). SARS-CoV-2 case and vaccination data are from 13 February 2020 to 30 November 2022.

	Bivariate Model					Full Multivariable Model					Simplified Multivariable Model				
	Coefficient	95% Confidence Interval		Marginal R^2^	Conditional R^2^	Coefficient	95% Confidence Interval		Marginal R^2^	Conditional R^2^	Coefficient	95% Confidence Interval		Marginal R^2^	Conditional R^2^
Null multi-level model	43.766	39.645	47.949	0.000	0.239										
Municipal level variables									0.523	0.581				0.496	0.594
% Mayan	−0.151	−0.206	−0.095	0.120	0.342	0.048	−0.008	0.105			-	-	-		
% Rural residence	−0.188	−0.251	−0.127	0.111	0.246	0.005	−0.061	0.068			-	-	-		
% Educational level primary school or above	1.400	1.184	1.616	0.400	0.451	0.894	0.534	1.259			0.736	0.383	1.075		
% Experiencing poverty	−0.542	−0.625	−0.460	0.405	0.462	−0.216	−0.409	−0.026			−0.249	−0.433	−0.070		
% Female sex	2.527	1.022	4.026	0.036	0.281	1.239	−0.210	2.662			-	-	-		
% in 0–17 age group	−1.813	−2.158	−1.470	0.296	0.388	0.999	0.296	1.714			1.065	0.355	1.767		
% in 60 or older age group	4.538	3.566	5.485	0.218	0.412	2.552	1.299	3.862			2.935	1.700	4.121		
% tested for SARS-CoV-2	0.521	0.431	0.613	0.314	0.460	0.215	0.085	0.344			0.246	0.137	0.355		
% died due to COVID-19	1.171	0.939	1.406	0.249	0.406	0.162	-0.148	0.463			-	-	-		
Departmental level variables															
Under-5 childhood mortality rate	−0.087	−0.461	0.268	0.003	0.249	0.199	−0.035	0.424			-	-	-		
% reporting difficulty accessing healthcare facilities due to distance	−0.822	−1.121	−0.532	0.176	0.249	−0.053	−0.447	0.338			-	-	-		
% 12–23 month olds receiving third Pentavalent vaccine	0.653	0.201	1.106	0.073	0.244	0.111	−0.273	0.496			-	-	-		
Gini coefficient	−0.619	−1.195	−0.084	0.052	0.245	−0.145	−0.594	0.298			-	-	-		

**Table 4 vaccines-11-00745-t004:** Association between sociodemographic factors (by municipalities and departments) and two-dose vaccination coverage (%) among patients aged 60 years or older by municipalities in Guatemala (N = 336). SARS-CoV-2 case and vaccination data are from 13 February 2020 to 30 November 2022.

	Bivariate Model					Full Multivariable Model					Simplified Multivariable Model				
	Coefficient	95% Confidence Interval		Marginal R^2^	Conditional R^2^	Coefficient	95% Confidence Interval		Marginal R^2^	Conditional R^2^	Coefficient	95% Confidence Interval		Marginal R^2^	Conditional R^2^
Null multi-level model	55.119	50.395	59.914	0.000	0.246										
Municipal level variables									0.477	0.615				0.487	0.600
% Mayan	−0.205	−0.268	−0.143	0.162	0.415	−0.016	−0.080	0.050			-	-	-		
% Rural residence	−0.181	−0.253	−0.112	0.079	0.241	0.097	0.027	0.165			0.094	0.027	0.159		
% Educational level primary school or above	1.649	1.394	1.903	0.409	0.513	0.965	0.558	1.377			0.978	0.566	1.372		
% Experiencing poverty	−0.638	−0.736	−0.542	0.411	0.526	−0.206	−0.392	−0.025			−0.201	−0.379	−0.028		
% Female sex	2.370	0.662	4.073	0.025	0.269	1.887	0.250	3.504			2.057	0.639	3.451		
% tested for SARS-CoV-2	0.595	0.492	0.699	0.310	0.484	0.301	0.151	0.451			0.314	0.190	0.437		
% died due to COVID-19	1.271	1.003	1.543	0.213	0.441	0.037	−0.318	0.385			-	-	-		
Departmental level variables															
Under-5 childhood mortality rate	−0.141	−0.568	0.263	0.006	0.256	0.241	−0.129	0.597			-	-	-		
% reporting difficulty accessing healthcare facilities due to distance	−0.824	−1.215	−0.445	0.138	0.251	0.265	−0.355	0.881			-	-	-		
% 12–23 month olds receiving third Pentavalent vaccine	0.641	0.099	1.181	0.055	0.252	0.147	−0.449	0.743			-	-	-		
Gini coefficient	−0.817	−1.447	−0.232	0.070	0.247	−0.475	−1.174	0.209			-	-	-		

**Table 5 vaccines-11-00745-t005:** Association between sociodemographic factors (by municipalities and departments) and two-dose vaccination coverage (%) by municipalities in Guatemala (N = 336). SARS-CoV-2 case and vaccination data are from 13 February 2020 to 1 October 2021.

	Bivariate Model					Full Multivariable Model					Simplified Multivariable Model				
	Coefficient	95% Confidence Interval		Marginal R^2^	Conditional R^2^	Coefficient	95% Confidence Interval		Marginal R^2^	Conditional R^2^	Coefficient	95% Confidence Interval		Marginal R^2^	Conditional R^2^
Null multi-level model	15.205	13.290	17.150	0.000	0.183										
Municipal level variables									0.550	0.615				0.546	0.615
% Mayan	−0.091	−0.118	−0.063	0.166	0.307	0.023	−0.005	0.051			-	-	-		
% Rural residence	−0.106	−0.138	−0.074	0.130	0.223	0.001	−0.032	0.032			-	-	-		
% Educational level primary school or above	0.715	0.601	0.829	0.384	0.438	0.230	0.049	0.413			0.161	−0.012	0.328		
% Experiencing poverty	−0.308	−0.350	−0.266	0.467	0.521	−0.174	−0.271	−0.079			−0.113	−0.190	−0.039		
% Female sex	1.039	0.246	1.829	0.023	0.225	−0.022	−0.752	0.695			-	-	-		
% in 0–17 age group	−1.094	−1.259	−0.931	0.395	0.441	0.332	−0.020	0.691			-	-	-		
% in 60 or older age group	2.832	2.351	3.301	0.315	0.450	1.679	1.053	2.335			1.393	0.893	1.873		
% tested for SARS-CoV-2	0.621	0.521	0.722	0.358	0.455	0.203	0.063	0.341			0.262	0.146	0.380		
% died due to COVID-19	0.784	0.646	0.923	0.304	0.397	0.140	-0.048	0.323			-	-	-		
Departmental level variables															
Under-5 childhood mortality rate	−0.056	−0.230	0.108	0.005	0.193	0.073	−0.051	0.192			-	-	-		
% reporting difficulty accessing healthcare facilities due to distance	−0.376	−0.518	−0.239	0.140	0.194	−0.011	−0.220	0.197			-	-	-		
% 12–23 month olds receiving third Pentavalent vaccine	0.290	0.080	0.501	0.056	0.180	−0.015	−0.218	0.188			-	-	-		
Gini coefficient	−0.137	−0.427	0.132	0.010	0.190	0.106	−0.132	0.339			-	-	-		

## Data Availability

Publicly available datasets were analyzed in this study. These data can be found here: Guatemala Population and Housing Census 2018 https://www.censopoblacion.gt/ (accessed on 5 December 2022); Guatemala Demographic and Health Survey https://dhsprogram.com/publications/publication-fr318-dhs-final-reports.cfm (accessed on 5 December 2022).; Ministry of Public Health and Social Assistance (MSPAS) of Guatemala https://tablerocovid.mspas.gob.gt/tablerocovid/ (accessed on 4 December 2022) and https://gtmvigilanciacovid.shinyapps.io/Coberturas_Tablero/ (accessed on 4 December 2022). Restrictions apply to the availability of the poverty indicator data. Data were obtained from Paolo Marsicovetere and are available from the author.

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
