# Peer review of "Sociodemographic Factors Associated with COVID-19 Vaccination among People in Guatemalan Municipalities"

_vaccines, 2023, doi:10.3390/vaccines11040745_

Round 1
Reviewer 1 Report
The topic of this study is extremely interesting. However, the method demonstrated in study lacks scientific rigors.
A detailed literature review is missing. Sociodemographic factors associated with COVID-19 vaccination among the people of Guatemalan municipalities might be new. However, the author needs to provide a possible list of previous studies that worked finding sociodemographic factors relevant to other countries / areas. Please portray the outcome of those existing studies in tabular manner and then demonstrate whether this study is contributing anything more or anything differently.
In this way, the contribution of this study to the body of knowledge would be clearer.
The Materials and Methods section doesn't portray the methodology in a comprehensive manner. Please add a flow chart or conceptual diagram for depicting the method more comprehensively.
A more systematic and scientific way of demonstrating the result of this study is by using one or more hypothesis. Articulate the research questions and hypothesis at the beginning of the paper (supported by existing literation) and then demonstrate with experimentation results whether null hypothesis is rejected or not (like https://www.mdpi.com/2079-9292/12/5/1205).
The separated figures of page 6, 7, 8 should be in a single page (i.e. Figure 1). Moreover, the quality of this figure 1, must be improved.
Table 3 is extremely difficult to read, because of overlapping and multi-line values. This table can be stretched across for making it readable. The same is issue present for Table 4 and Table 5.
Author Response
The topic of this study is extremely interesting. However, the method demonstrated in study lacks scientific rigors. A detailed literature review is missing. Sociodemographic factors associated with COVID-19 vaccination among the people of Guatemalan municipalities might be new. However, the author needs to provide a possible list of previous studies that worked finding sociodemographic factors relevant to other countries / areas. Please portray the outcome of those existing studies in tabular manner and then demonstrate whether this study is contributing anything more or anything differently. In this way, the contribution of this study to the body of knowledge would be clearer.
Thank you for your helpful comments on improving the background provided in our manuscript. We agree that adding more literature on what is known on sociodemographic factors associated with COVID-19 vaccine uptake would be helpful to readers. Unfortunately, much of the literature we have found has been focused on disparities within the United States or is focused on vaccination intention rather than uptake. We’ve added a paragraph in the introduction citing the three studies below that discuss factors associated with lower COVID-19 vaccination within the United States, such as rurality, lower income, and lower educational achievement (lines 65-75). Much of the other studies we have found in Latin American and the Caribbean focus on intent to vaccinate. We feel that adding a table of studies would be beyond the scope of this analysis, as we are not conducting a systematic review of the literature. Additionally, our focus is on Guatemala, as we hope to provide information on intra-country disparities that can be used for future investigations or public health response.
Baack BN, Abad N, Yankey D, et al. COVID-19 Vaccination Coverage and Intent Among Adults Aged 18-39 Years - United States, March-May 2021. MMWR Morb Mortal Wkly Rep. 2021;70(25):928-933.
Murthy BP, Sterrett N, Weller D, et al. Disparities in COVID-19 Vaccination Coverage Between Urban and Rural Counties - United States, December 14, 2020-April 10, 2021. MMWR Morb Mortal Wkly Rep. 2021;70(20):759-764.
Hughes MM, Wang A, Grossman MK, et al. County-Level COVID-19 Vaccination Coverage and Social Vulnerability - United States, December 14, 2020-March 1, 2021. MMWR Morb Mortal Wkly Rep. 2021;70(12):431-436.
The Materials and Methods section doesn't portray the methodology in a comprehensive manner. Please add a flow chart or conceptual diagram for depicting the method more comprehensively.
Thank you, and I hope we can provide some clarification on our methodology. In our analysis, we used multivariable modeling to assess associations between sociodemographic variables and COVID-19 vaccination. Table 1 describes the variables included in the model and Tables 3-5 show the results of the models we constructed. We do not think that a flow-chart or diagram would provide more information as the tables we have constructed already show which variables were included in the bivariate, full multivariable and simplified multivariable models. Further, now that we have improved the formatting of Tables 3-5, we think that the methodology is more clear.
A more systematic and scientific way of demonstrating the result of this study is by using one or more hypothesis. Articulate the research questions and hypothesis at the beginning of the paper (supported by existing literation) and then demonstrate with experimentation results whether null hypothesis is rejected or not (like https://www.mdpi.com/2079-9292/12/5/1205).
As described in the Introduction, our research question was focused on identifying factors associated with COVID-19 vaccination at the municipality level in Guatemala. Based on our literature review we hypothesized that COVID-19 vaccination would be lower in areas with higher poverty, rurality, and indigenous groups. We have added this hypothesis statement at the end of our Methods section: “We hypothesized based on our literature review that COVID-19 vaccination would be negatively associated with higher poverty, rurality, and indigenousness" (lines 182-183). As our analysis was focused on identifying these factors rather than proving or disproving a hypothesis, we feel it would be beyond the scope of this analysis to add hypothesis testing to this manuscript.
The separated figures of page 6, 7, 8 should be in a single page (i.e. Figure 1). Moreover, the quality of this figure 1, must be improved.
Thank you, we have merged the three parts of Figure 1 onto one page and improved the clarity of the image. The editorial staff may let us know if the figure is too small for readers in this format.
Table 3 is extremely difficult to read, because of overlapping and multi-line values. This table can be stretched across for making it readable. The same is issue present for Table 4 and Table 5.
Our intent was for Tables 3-5 to be in landscape orientation though the submission system may not allow for that format. We will discuss with the editorial staff if landscape orientation is possible for better readability and improve the formatting.
Reviewer 2 Report
Dear authors
I am delighted to read the manuscript.
I have reviewed the manuscript and I believe that no more edits are needed. It is an ecological study which of course has some limitations (which was mentioned in the limitation) but provides information at country level. Results of such studies are beneficial for policy makers when deciding on community health interventions.
If possible please provide data on cost of vaccination for individuals.
Author Response
Dear authors, I am delighted to read the manuscript. I have reviewed the manuscript and I believe that no more edits are needed. It is an ecological study which of course has some limitations (which was mentioned in the limitation) but provides information at country level. Results of such studies are beneficial for policy makers when deciding on community health interventions.
If possible please provide data on cost of vaccination for individuals.
Thank you for your comments. Vaccines were provided at no cost to Guatemalans through the Ministry of Public Health and Social Assistance and we added the phrase “free of charge” to our introduction to clarify this (line 50). We agree that other costs, such as monetary and opportunity costs of accessing vaccination sites, missing work, arranging childcare, etc, would be beneficial to include in our discussion, however, we are unable to estimate these costs at this time. This would be an excellent area of future work and we have added the sentence “Additional research on monetary and opportunity costs of accessing vaccination within Guatemala may be needed” in our discussion reiterating this observation (lines 464-466).
Round 2
Reviewer 1 Report
The authors have responded to my previous queries appropriately. They also updated the manuscript accordingly. I have no further comments to add.